# Green Machining of NFRP Material

**Zuzana Mitaľová** [1,*] , **Dagmar Klichová** [2] , **František Botko** [1] , **Juliána Litecká** [3] , **Radoslav Vandžura** [1] and **Dušan Mitaľ** [1]

1  Faculty of Manufacturing Technologies, Technical University of Košice, 080 01 Prešov, Slovakia; frantisek.botko@tuke.sk (F.B.); radoslav.vandzura@tuke.sk (R.V.); dusan.mital@tuke.sk (D.M.)
2  The Czech Academy of Sciences, Institute of Geonics, 708 00 Ostrava, Czech Republic; dagmar.klichova@ugn.cas.cz
3  Faculty of Humanities and Natural Science, University of Prešov, 080 01 Prešov, Slovakia; juliana.litecka@unipo.sk
*  Correspondence: zuzana.mitalova@tuke.sk; Tel.: +421-55-602-6465

**Abstract:** Nowadays, great emphasis is placed on environmental aspects of production processes with focus to lower carbon footprint. Natural fibre-reinforced plastics (NFRP) show potential for application in many fields of industry due their specific properties. Machining of NFRP-based materials is meeting several problems arising from non-homogenous structure as well as plastic-based matrix. Machining of NFRP using conventional technologies meets limitations due to the properties and geometry of the tools. Abrasive water jet (AWJ) machining can solve some of the problems machining NFRP materials. The presented article focused on surface topography evaluation of one kind of NFRP composite material after cutting by AWJ. Optical profilometry and 3D microscopy were applied for measurement of surface roughness parameters of surfaces created by AWJ with variable cutting parameters. Maximal height of profile $Rz$ was measured in 20 lines perpendicular to the jet direction form upper to lower cut line. Structure of cut surface was observed and evaluated for different technologic parameters. The obtained results show promising presuppositions for application of AWJ technology for cutting of NFRP based materials.

**Keywords:** water jet cutting; non-conventional machining; NFRP composite; natural fibre-reinforced plastics; wood plastic composite; WPC





## 1. Introduction

The current interest in the use of natural materials as a substitute for fossil raw materials is justified from a material, technological, financial and, last but not least, environmental point of view (reduction in carbon footprint—$CO_2$). The trend was long supported by the European Commission (the *European Green Deal* programme for minimising the carbon footprint and reducing emissions in the long term). In the last decade, NFRP (natural fibre-reinforced polymer) materials replaced several composite materials reinforced with synthetic fibres, the recycling of which is difficult (e.g., glass fibres—for comparison of the characteristics of synthetic versus natural fibres—see Table 1). NFRP composite materials contain natural fibres and a polymer matrix in a specified ratio (+ additional substances—additives). Thermoplastics materials usually dominate in matrices for NFRP: polypropylene (PP), polyethylene (PE), polyvinylchloride (PVC), polyamide (PA). On the other hand, thermoset materials like phenolic, epoxy and polyester are the most commonly used matrices [1].

The lignocellulosic fibres in the polymer matrix are obtained from cotton, loofah, sisal, flax, hemp, ramie, coconut, bamboo, banana, pineapple, palm leaves, rice, maize, barley husks and deciduous/coniferous trees (Figure 1 shows the chemical composition of the lignocellulosic fibres—consist of lignin, hemicellulose, cellulose). Their advantages include non-toxicity, relatively high strength, low weight/low price independent of the oil price

and biodegradability. The resulting NFRP properties of the composite material depend on the percentage (%) in which the of individual components account for the total volume, their interfacial adhesion, production technology, fibre origin or production/finishing technologies.

**Table 1.** Comparison of selected characteristics of synthetic and natural fibres [2].

| Characteristics | Natural Fibres (NF) | Synthetic Fibres (e.g., Glass Fibre) |
|---|---|---|
| Cost | ↓ Low | ↑ higher compared to NF |
| Density | ↓ Low | ↑ double compared NF |
| Recyclability | ✓ Yes | ✘ No |
| Renewability | ✓ Yes | ✘ No |
| $CO_2$ neutrality | ✓ Yes | ✘ No |
| Abrasion to machines | ✘ No | ✓ Yes |
| Health risk associated with production | ✘ No | ✓ Yes |
| Disposal | Biodegradable | Non-biodegradable |

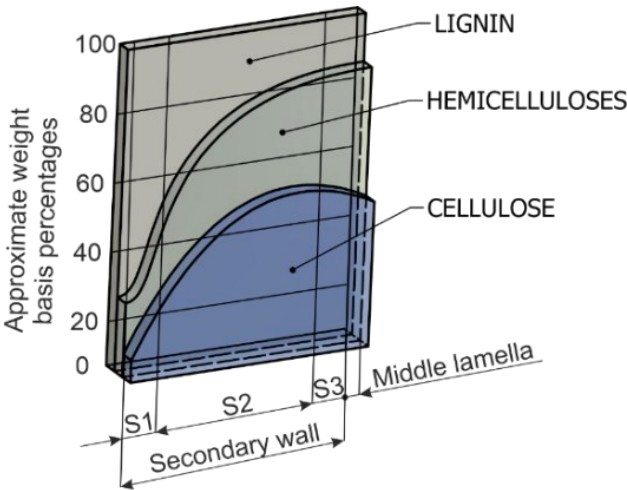

**Figure 1.** Chemical composition of a typical lignocellulosic fibre (lignin + hemicellulose + cellulose in percentual ratio) [3].

The first NFRP composite material for manufacture of vast numbers of boards/tubes for electronic purpose was used as early as in 1908 (sheet from phenol/formaldehyde resins reinforced with paper or cotton). Due to a reduction in vehicle weight, $CO_2$ emissions and price, NFRP composite materials were also applied in the automotive sector. In 2012, more than 124,000 tonnes of natural fibre composites were shipped globally for automotive purposes. The available research shows that the use of selected types of natural fibres increased the mechanical properties of the resulting parts (jute, hemp, flax, banana and bamboo fibres—the mechanical properties listed in Table 2) [4,5].

**Table 2.** Mechanical properties of selected natural fibres [6–9].

| Fibre Type [1] | Tensile Strength [MPa] | Tensibility [%] | Young's Modulus [GPa] | Study |
|---|---|---|---|---|
| Jute fibre | 393–773 | 1.5–1.8 | 26.5 | Kumar and Sharma, 2007 [6] |
| Hemp fibre | 550–900 | 2.0–3.0 | 70 | Lu et al., 2012 [7] |
| Flax fibre | 800–1500 | 2.7–3.2 | 60–80 | Shashria, 2019 [8] |
| Banana fibre | 540–900 | - | 34.8 | Narayanan and Elazaperumal, 2012 [9] |

[1] Values may vary depending on the type and origin of the fibres.

The composite retains its toughness and is suitable for visually as well as structurally functional parts—e.g., door panels, columns, consoles, etc. Adding wood fibres (particles in the form of flour) to PE/PP/PVC yields a material applied in the construction industry—wood plastic composite (in short, WPC).

As a filler, the composite is typically used in wood plastic composite product, a wood flour obtained from maple, pine, cedar, spruce-fir or oak (determined by geographical location, type, price and availability). The composite product subjected to extreme moisture can be applied to obtain a flour extract from red maple—which improves water resistance of used composite material. Or, to improve durability, it is possible to use a resin of Guayele plants: *Parthenium argentatum* (normally obtained as an additional product when processing rubber). Hardwood provides better tensile properties and heat deflection when compared with softwood. The wood flour presents the crushed wood grains and there is appearance similar to a standard flour—this is a term commonly used in the practice. There are two steps in wood production: first step—size reduction (using a hammer mill/attrition mill or chipper) and second step—a size classification (screening by sieves). In the process of wood flour production, it is necessary to include the drying phase. The usual content of water in wood fibre/flour is about 5–15%, where increasing humidity reduces mechanical properties and thermal stability of the final composite product. For wood flour, it is necessary to keep the moisture content below 1% (in relation to this fact, it is requisite to storage flour in closed plastic bags, until mixing with polymer). The particles are sorted by a size—using sieves with different mesh—and are then classified according to US standard (for example: a particle with a diameter of 850 μm with an identification of 20 MESH/a particle with a diameter of 250 μm with an identification of 60 MESH according to the standard). In the practice, flour with a size between 60 and 80 MESH can be used. For WPC production, it is possible to apply waste from the woodworking industry, too. For example: a co-product like trimming for sawmills, breakdown of urban and demolition wood, or logging trimmings/slash. Adding starch particle into a matrix helps to prevent the degradation (starches from potatoes, rice, grain from cereals). During the melting process, it is necessary to ensure that the temperature does not exceed 200 °C, though some studies state 180 °C, because it would cause wood decomposition. Due to the thermal stability of wood, thermoplastics are used because they can be processed at relatively low temperatures below wood's thermal degradation. Thermoplastics are polymers, which turn to a liquid when heated and a freezy-to-glassy state when cooled. High density polyethylene (HDPE), polypropylene (PP) and polyvinylchloride (PVC) are the most common thermoplastic polymers used in WPCs (according to Klyosov, 2007). Currently studies are pointing to the application of polymethyl methacrylate (PMMA) or Nylon 6. Similarly, biopolymers and biodegradable polymers can be applied as an adequate substitute for thermoplastics (PLA—polylactic acid, or based on polyhydroxyalkanoates—PHB/PHBV). PLA and PHB matrices have similar properties to polypropylene matrices. Additives improve mechanical properties of a final product, provide a chemical stability and more easily process (coupling agents, lubricants, stabilizers, flame retardants, biocides, pigments, fillers, chemical and physical blowing agents) [1,10–14].

The final properties of WPC products depend on several attributes:

- Type of applied polymer matrix;
- The percentage of organic reinforcement (type of wood or plant fibres), the morphology of particles, their physical properties and moisture content;
- Percentage of individual additives;
- Technology and conditions of production process;
- The origin of raw components (possibility of applied plastic recycled, materials, geographical location, etc.);
- Interaction between components.

General WPC production processes are [1,10]:

- Extrusion—for linear profile (technology applied in up to 97% of manufactured WPC products);

- Injection moulding—for 3D parts of regular or irregular shapes, application in mass production, suitable for polymers with low molecular weight. The production process is fundamentally like the production of injected plastics, (disadvantages of the technology: high procurement costs, imperfect compounding of the components of WPC composite material/advantages: minimal waste, short cycle time, observe required dimensional tolerances);
- Compression moulding—cost-effective production of complex parts (disadvantages of technology: complex mould design, the ratio of applied resin and fibre is difficult to control, advantages: no need of qualified staff);
- Calandering—special way of "rolling" in floor production.

Applying thermoplastic matrix offering usage of FDM technology (fused deposition modelling), using direct layering of melted material via heated 3D print head.

WPC composites are applied to flooring, outdoor bed boards, park benches, building templates, production of pallets, fencing and the automotive industry (especially in China). European and American companies recently saw an increase in consumer demand for these building materials. The European WPC market was estimated at over 450 kilotons in 2021 [1,10,15].

The European market is dominated by Tecnaro GMBH, Jelu-werk J. Ehler gmbH & Co. KG, Novo-Tech Trading GmbH & Co. KG. Currently, NFRP composites are used for packing, too. For example, ENKEV produced a product such as Cocolok made from natural fibres/coconut and latex, or applied starch-based composite for tableware. NFRP composite materials are also used in the production of canoes, sporting goods, musical instruments and the transport industry, too [16,17].

Conventional NFRP machining is a long-lasting problem. The existing literature demonstrated that the machinability of composite materials is not the same the machining of homogeneous materials (compared to conventional metal working) [18].

The reasons for this are: molecular nature of the composite material, its inhomogeneity (heterogeneity), fibres pull out during drilling operations, low production productivity, poor technological inheritance indicators values—for example, surface roughness. The resulting characteristics (*Rz*—maximum height and *Ra*—average roughness) depend on the variable conditions of the production technology and the geometry of the applied tool [19–23].

The possibility of resolving the problem is by machining by the AWJ technology. The technology itself offers the following advantages: absence of heat-affected zone (HAZ), no potential fire hazard, low stress on the workpiece, higher flexibility and productivity and no aerosol generation; hence, AWJ machining is called green machining. Machining of NFRP composite materials using the AWJ technology depends on several process variables such as hydraulic (water) pressure, type of the NFRP material, nozzle distance, abrasive type/abrasive size, mass flow rate, etc. The output parameters (surface roughness, material removal rate, kerf taper angle) of the AWJ cutting process are related to the input parameters [24–27].

Researchers examined machining of "green composites" by AWJ/WJ, but very few of them addressed machining of the NFRP composite materials filled with wood. One of the latest studies is the work of Boopathi et al. (2022), dealing with the impact of the input parameters traverse speed and water jet pressure on the output characteristics—surface roughness SR and kerf angle KA (using the Taguchi method), when machining a composite material with neem wood saw powder with PP (polypropylene matrix). They observed that SR and KA were greatly impacted by the percentage of neem wood saw particles, traverse speed and water jet pressure. Predicting the optimal surface roughness after abrasive water jet machining of the NFRP material (fill: the sundi wood dust) was the topic of the work of Jagadish et al. (2019). Based on the results and the methods applied (fuzzy logic and regression analysis), optimum input parameters and the fundamental impact of AMGS (abrasive material grain size) and AMFR (abrasive mass flow rate) on the resulting surface were evaluated. It also mentioned a finding that a significant amount of

crack propagation, some voids on the machined surface, were found due to the presence of moisture content in the natural fillers. The authors Jagadish and Gupta also processed the results in the form of a book publication: Abrasive Water Jet Machining of Engineering Materials, extended to the machining of metallic materials, polymer composites and ceramic composites. Hutyrová et al. (2016) dealt with verification of suitability of water jet/abrasive water jet application to disintegration of turning NFRP samples (wood particles + HDPE matrix). Surface quality was investigated by optical profilometry and the AWJ machining technology was evaluated as a suitable alternative to machining, subject to setting optimal process parameters [24,28–30].

This paper dealt with the evaluation of the characteristics of technological inheritance—namely, profile parameters of surface roughness (*Rz*) in relation to the changing input parameters of the AWJ cutting process. Surface quality of machined surfaces was analysed using an optical profilometer MicroProf FRT, and the macrostructure of the created surface was visualized using the VHX-6000 digital microscope at 50× magnification (Keyence, Osaka, Japan).

## 2. Materials and Methods

Material: in the experiment, samples with dimensions of 40 × 60 × 1500 mm were used (technology of the profile production: process of extrusion). The NFRP composite material consisted of high-density PE matrix filled with wood reinforcement in a ratio of 25/75 vol.% (+additives). The mechanical properties of the examined samples are expressed in the graphical display—Figure 2.

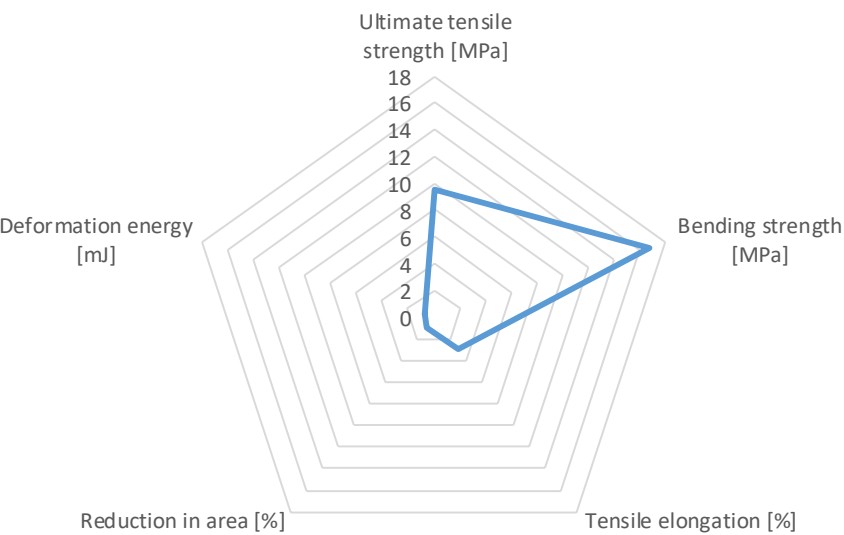

**Figure 2.** Mechanical properties of the examined wood plastic composite material (values are statistically verified).

Water Jet 3015 RT-3D was used for the AWJ cutting of the samples investigated. Required water pressure was supplied by the PTV JETS—3.8/60 Classic pump. Abrasive particles (Australian garnet, 80 MESH, were fed from a hopper through an abrasive management system by a tube with the inner diameter of 6.4 mm with the calibrated dosing accuracy of ±2.0 g. The two factors varied in course of the experiment (in relation to the controlled result parameter Q):

- Abrasive mass flow rate $m_a$ [g·min$^{-1}$];
- Traverse speed of cutting head $v_f$ [mm·min$^{-1}$].

The values of selected cutting process parameters (including sample designation) and machine settings are given in Tables 3 and 4.

**Table 3.** Design of sample testing and values of variable factors of the cutting process by AWJ.

| Sample No. | Abrasive Mass Flow Rate $m_a$ [g·min$^{-1}$] | Traverse Speed $v_f$ [mm·min$^{-1}$] |
|---|---|---|
| S1 | 150 | 346 |
| S2 | 200 | 387 |
| S3 | 250 | 423 |
| S4 | 300 | 455 |
| ⌃ | . | Q5 |
| ⋮ | . | ⋮ |
| | . | |
| Q1 | . | ⌄ |
| S17 | 150 | 80 |
| S18 | 200 | 90 |
| S19 | 250 | 100 |
| S20 | 300 | 105 |

The machined surfaces were scanned by the MicroProf FRT profilometer (working based on chromatic aberration), and the basic surface roughness characteristics were determined in accordance with ISO 21920-2, ISO 16610-21 (filtration of the selected parameters took place in accordance with the standard) [31,32]. The measurement was carried out in 20 lines along the sample, marked h1–h39, in direction perpendicular to direction of the water jet impact (Figure 3). Total length evaluated: 40 mm. Position of the last line < 2 mm from the lower edge—value determined by the SN 214001:2010 standard [33]. Figures 4 and 5 represents the lines which are considered as a transition between areas with different roughness.

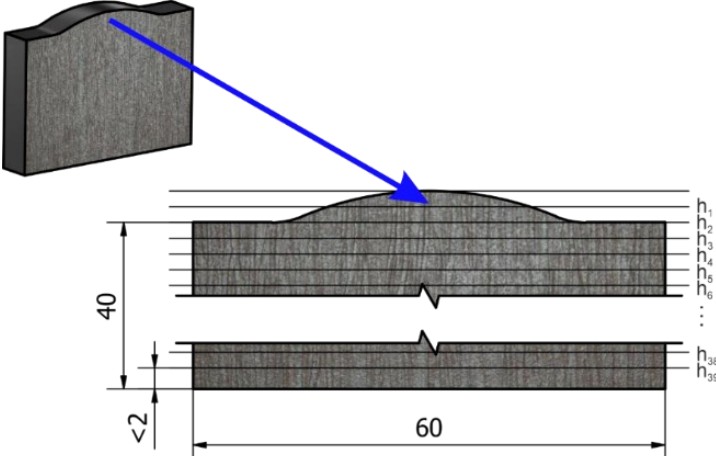

**Figure 3.** Positioning of lines along area under evaluation (last line least 2 mm from the bottom edge of the sample).

**Table 4.** Parameters of AWJ process cutting.

| Parameters | Value | Unit |
|---|---|---|
| Water pressure $p$ | 400 | MPa |
| Water orifice diameter $d_0$ | 0.3 | mm |
| Focusing tube diameter $d_f$ | 0.9 | mm |
| Standoff distance | 4 | mm |
| Abrasive mass flow rate $m_a$ | 150–300 | g·min$^{-1}$ |
| Traverse speed $v_f$ | Variable [2] | mm·min$^{-1}$ |
| Abrasive material | - | Australian Garnet |
| Abrasive particle | 80 | MESH |

[2] Based on required cut quality Q1 to Q5 (set by the SN 214001:2010 standard [33]).

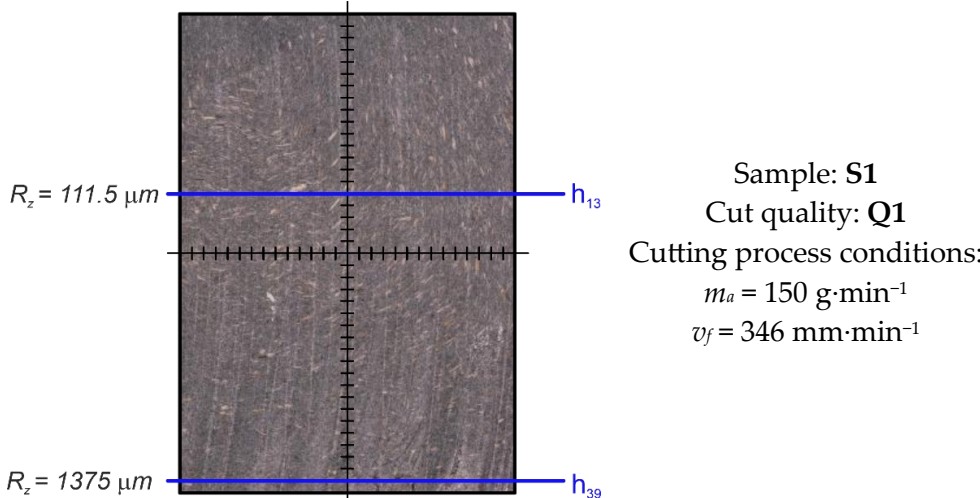

**Figure 4.** Selected lines h13/h39—measured of the *Rz* parameter in the lines defined (sample: S1, cut quality: Q1).

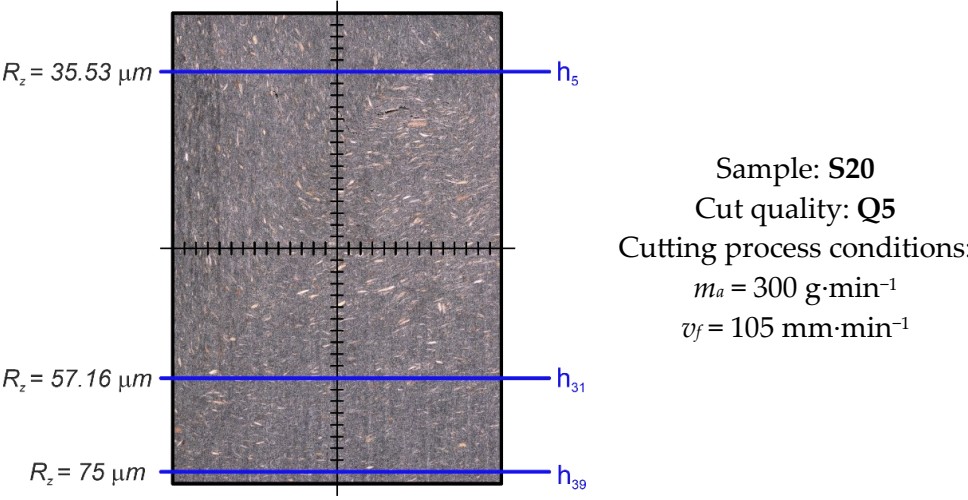

**Figure 5.** Selected lines h5/h31/h39—measured of the *Rz* parameter in the lines defined (sample: S20, cut quality: Q5).

Once the experimental measurements were performed and the measured data processed (by the Mountains SPIP 6.7.7 Academic software), information about the size of the surface topography unevenness dependent on the changing experimental technological parameters was obtained. In relation to the resulting Q parameter—it was necessary to evaluate individual sets separately. Based on SN 214001: 2010 (Contact-Free Cutting—Water Jet Cutting—Geometrical Product Specification and Quality from SAI Global,), the surfaces were categorized as Q1 to Q5, where the Q1 cut quality was defined as the lowest/Q5 cut quality as the highest [33].

## 3. Results and Discussion

In relation to the increasing trend of values, line charts were selected to display the Q1 and Q5 cut quality sets (making it possible to predict the *Rz* values between the lines recorded). Subsequently, the least squares method was used to construct the trend lines (regression equations—Table 5) of exponential dependencies (including the confidence factors) for the Q1 set. Based on the regression analysis of measured data, regression equations were created, which described dependence of maximal height of the profile (*Rz*) on the distance from the upper cut line. Since the trend of dependent variable was

nonlinear, exponential equations were created after analysis. Exponential equation in basic format is represented by prescription:

$$Y = \beta_0 \cdot X^{\beta_1} + \varepsilon$$

where $\beta_0$ and $\beta_1$ are the model parameters and $\varepsilon$ is the random observation error.

**Table 5.** Equations of exponential dependencies of the SET Q1.

| Sample No. | Equation of the *Rz* Dependence in Relation to the Depth Line |
|:---:|:---:|
| S1 | $y = 28.304e^{0.1914x} / R^2 = 0.9578$ |
| S2 | $y = 22.113e^{0.1931x} / R^2 = 0.9151$ |
| S3 | $y = 36.013e^{0.1748x} / R^2 = 0.9838$ |
| S4 | $y = 39.418e^{0.1529x} / R^2 = 0.9756$ |

Functional dependencies are considered statistically significant where $R^2 > 0.9$ (if the $\rightarrow$ exponential equation is $R^2 = 0$ not suitable for assuming the behaviour of values / $R^2 = 1$ $\rightarrow$ there is a perfect correlation between the actual values and the values specified by the trendline). In the case of Q1 set, all dependencies described were statistically significant. Based on microscopic observation, the surfaces of the Q1 set samples can be divided into 2 different zones: a smooth zone and a coarse zone. Significant plastic deformation of the surface was recorded starting from the line h13. The highest value of the *Rz* parameter measured was recorded in the sample S3 in the h39 line, of 1536.0 µm. With the increasing cut depth, the *Rz* parameter values increased exponentially (see Figure 6). The exponential trend of observed parameters can be explained, since, in the process of abrasive water jet cutting, loss of the kinetic energy of the water jet and abrasive particles occurred via lowering the velocity in the direction from upper to lower cut line.

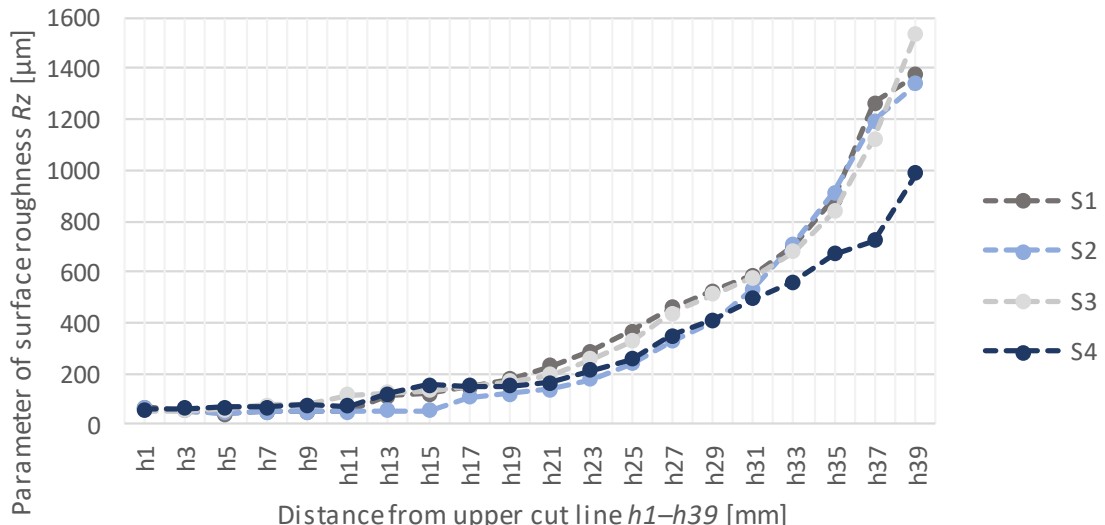

**Figure 6.** The course of the tallest height of *Rz* unevenness in lines h1–h39 (Q1 set samples).

In relation to the nature of the course of *Rz* parameters of the profile surface roughness, no regression dependencies for Q5 set were made (Figure 7). Samples of the Q5 set were divided into three zones: initiation, smooth (h5–h31) and coarse. The width of the initiation zone was defined by local minima in the h5 line (see Table 6). Within the defined smooth zone, the profile surface roughness parameters *Rz* ranged from 36.90 µm to 71.89 µm.

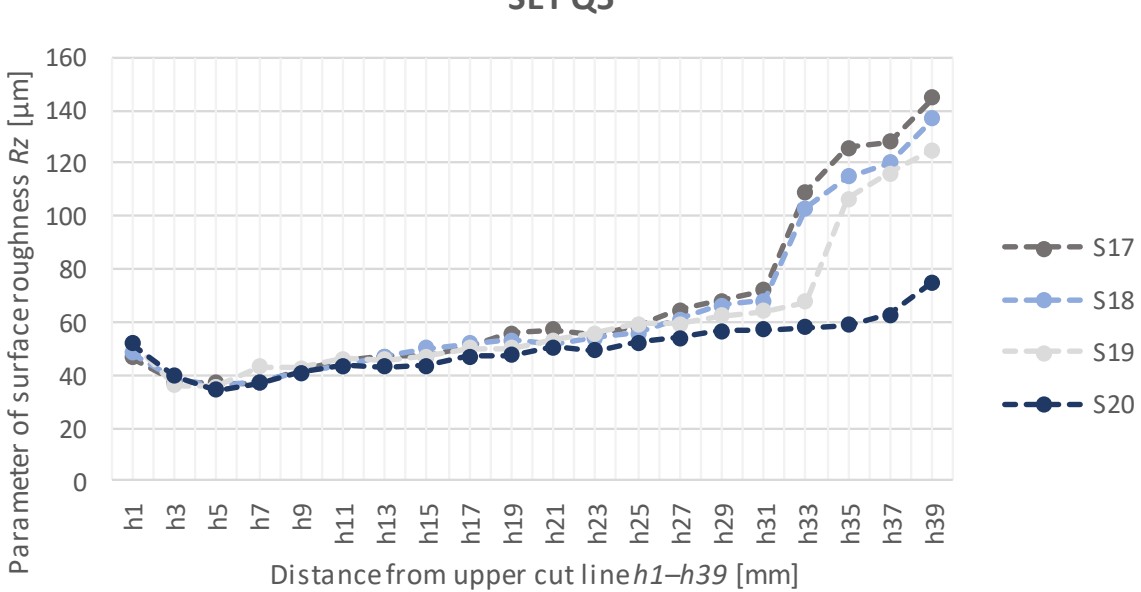

**Figure 7.** The course of the tallest height of *Rz* unevenness in lines h1–h39 (Q5 set samples).

**Table 6.** *Rz* parameter values in defined lines (defining transitions between individual zones).

| Line Number hx | Sample S1 (Cut Quality: Q1) | Sample S2 (Cut Quality: Q1) | Sample S3 (Cut Quality: Q1) | Sample S4 (Cut Quality: Q1) |
|---|---|---|---|---|
| h13 | 111.5 μm | 53.05 μm | 123.50 μm | 120.60 μm |
| h39 | 1375.0 μm | 1341.0 μm | 1536.0 μm | 989.7 μm |
| **Line Number hx** | **Sample S17 (Cut Quality: Q5)** | **Sample S18 (Cut Quality: Q5)** | **Sample S19 (Cut Quality: Q5)** | **Sample S20 (Cut Quality: Q5)** |
| h5 | 37.00 μm | 35.80 μm | 35.67 μm | 34.53 μm |
| h31 | 71.89 μm | 68.14 μm | 63.93 μm | 57.16 μm |
| h39 | 144.70 μm | 136.70 μm | 125.00 μm | 75.00 μm |

The resulting surfaces of sets Q1 and Q5, and the associated unevenness, showed a specific morphology due to cutting under different technological parameters. Based on the shape on the lower cut line at the outlet of the water jet, it was possible to assess the "suitability" of the choice of technological parameters in relation to the physical and mechanical properties of the material [34].

The conditions proposed for set Q1 and set Q5 confirmed the previous claim. It was evident that with the decreasing traverse speed (and the increasing abrasive mass flow), the cut surfaces exhibited deformations that were less pronounced (Table 7, Figure 8). In Table 7, surfaces created by a defined combination of factors, abrasive mass flow rate $m_a$/traverse speed $v_f$, are visualized.

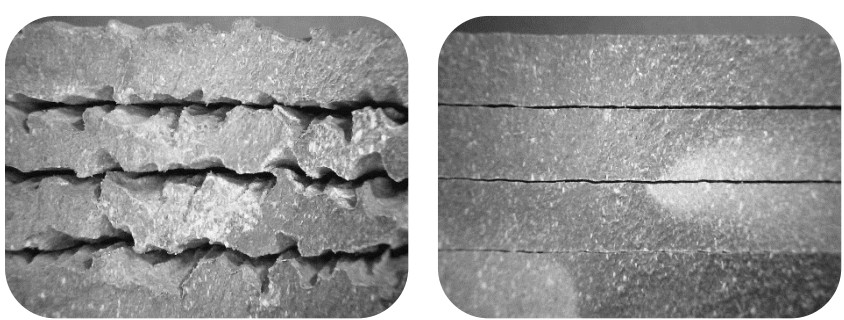

**Figure 8.** Cut at water jet outlet (left: Q1 set samples/right: Q5 set samples).

**Table 7.** Images of sample surface structure taken with the VHX-6000 digital microscope. Samples: S1–S4 (Q1), S17–S20 (Q5).

| S1<br>(Cut Quality: Q1) | S2<br>(Cut Quality: Q1) | S3<br>(Cut Quality: Q1) | S4<br>(Cut Quality: Q1) |
|---|---|---|---|
| S17<br>(Cut Quality:Q5) | S18<br>(Cut Quality: Q5) | S19<br>(Cut Quality: Q5) | S20<br>(Cut Quality: Q5) |

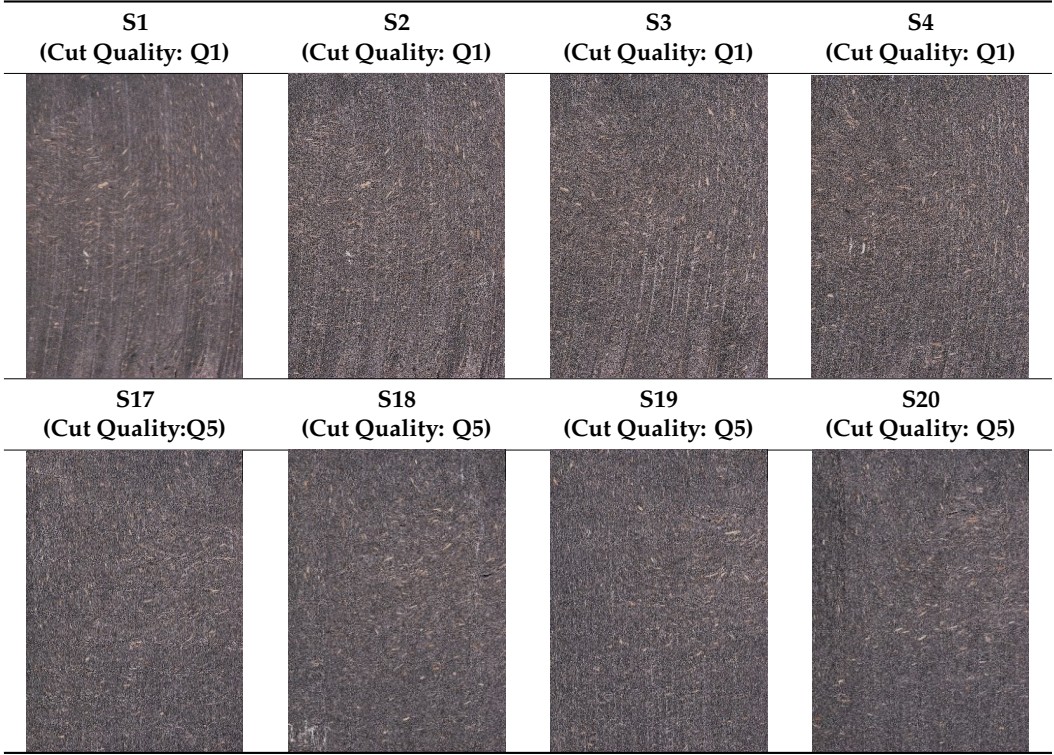

The macrostructure of the surface was visualized using the VHX-6000 digital microscope (Keyence) at 50× magnification. The surface area of 2.6 × 3.6 mm was scanned by the stapling method. In relation to the technology of profile production—by extrusion—the orientation of wood fibres and particles followed the flow of polymer. The photographed area (detail Figure 9) showed wood particles with different colours and fractions. At the interface of the components: wood versus plastic, cracks were visible that were created by imperfect encapsulation of wood particles (the cause of imperfect encapsulation is the low adhesion of wood particles to resins/matrices, the complex shape of the surface). Based on the analysis of samples from 1 to 20, their size ranged from a few micrometres to about 4 mm. The most pronounced cracks were on the samples designated 18 and 20. The mechanism of tracks (grooves/furrows) formation was given by the trajectory of movement and the effect of abrasive particles on the surface of the cut (Figure 10).

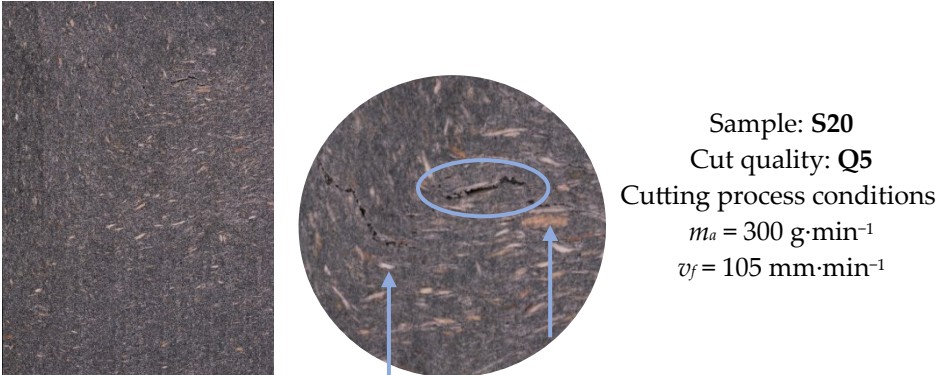

Sample: **S20**
Cut quality: **Q5**
Cutting process conditions:
$m_a$ = 300 g·min$^{-1}$
$v_f$ = 105 mm·min$^{-1}$

**Figure 9.** Significant cracks at the top of the profile cut by AWJ (arrows are pointing on natural reinforcement with different fractions and colours).

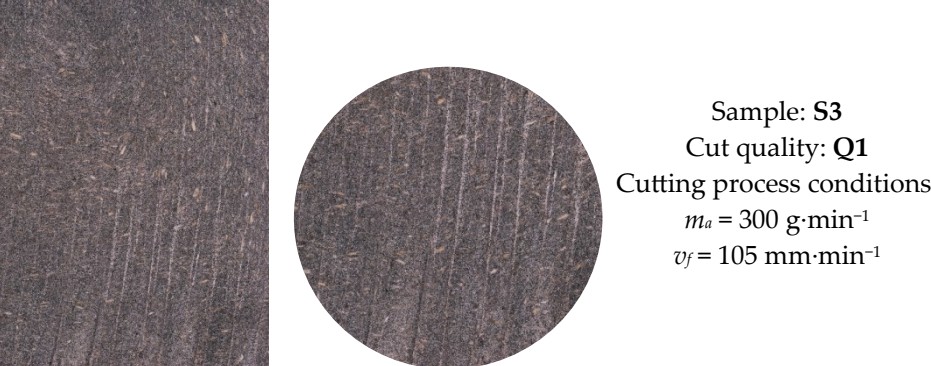

Sample: **S3**
Cut quality: **Q1**
Cutting process conditions:
$m_a$ = 300 g·min⁻¹
$v_f$ = 105 mm·min⁻¹

**Figure 10.** Pronounced furrows (tracks) left by AWJ in the final area of the cut.

## 4. Conclusions

Conventional machining of the composite materials with natural fibres is a long-lasting problem (compared to conventional metal working). The application of the AWJ technology eliminates the problem of tool wear, in the sense that the plastic matrix partially melts and sticks to the functional surfaces of the tool used. Surface topography was investigated by optical profilometry and macroscopic evaluation. Sets of samples with defined cut quality were analysed: Q1 (lowest cut quality) and Q5 (highest cut quality). The cutting parameters were selected with respect to the required quality Q (values of the ma parameter: selected/values of the $v_f$ parameter: calculated by the machine control system with respect to the expected Q, based on the material library). It was evident that with the decreasing traverse speed of the cutting head (and increasing mass flow of the abrasive), the cut surfaces exhibited less pronounced deformations. In addition to different technological parameters of the sets, the resulting surface quality also depended on the physical properties of the material's NFRP components. Subject to optimal cutting process parameters, the AWJ technology can be applied as a suitable alternative for cutting the NFRP materials (considering the mechanical properties of the composite, the type and orientation of the fibres/particles).

**Author Contributions:** Conceptualization, Z.M.; methodology, D.K. and Z.M.; software, F.B. and R.V.; investigation, Z.M. and D.M.; resources, F.B.; data curation, D.K. and Z.M.; writing—original draft preparation, F.B. and Z.M.; writing—review and editing, F.B.; visualization, J.L.; supervision, Z.M. and F.B. All authors have read and agreed to the published version of the manuscript.

**Funding:** This work was supported by the Slovak Research and Development Agency under the contract No. APVV-20-0514 (Research of the influence of technological parameters of abrasive water jet machining on the surface integrity of tool steels). Research was supported by project KEGA no. 032TUKE4/2021 Transfer and implementation of knowledge in the field of energy beam technologies into study programs of technical secondary schools supporting dual education and project VEGA 1/0080/20 Research into the effect of high speed and high feed machining technologies on the surface integrity of hard-to-machine materials.

**Data Availability Statement:** Not applicable.

**Conflicts of Interest:** The authors declare no conflict of interest.

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
