# Peer review of "Green Machining of NFRP Material"

_machines, doi:10.3390/machines11070692_

Round 1

Reviewer 1 Report

The authors presented surface topography evaluation of NFRP material after cutting by AWJ.  The paper is focused on optical profilometry, and 3D microscopy was applied for measurement of surface roughness parameters of surfaces created by AWJ with variable cutting parameters. In the author's opinion the obtained results show promising presuppositions for application of AWJ technology for cutting of NFRP based materials.

The paper is remarkably interesting, and I haven't any fundamental objections. In my opinion, it would only be appropriate to include a slightly broader analysis of the state of the issue related to surface quality with the issues on the use of fractal analysis for evaluating the surface quality generated by waterjet.

The paper is written clearly and lucidly. The layout is correct, the order of the chapters is good. Conclusions adequate to the content.

Author Response

Good morning,

Thank you very much for advice.

Publications related to machining of WPC materials by AWJ technology are not available, so has been added further information about WPC into the introduction.

Best regards

Zuzana Mitalova

Reviewer 2 Report

This is an interesting article concerning  machining of NFRP material. The authors clearly determined the aim of the study. The introduction presents the scientific problem in a comprehensive manner. The paper is well written - clear motivation, explanations. The drawings are of good quality. However, before allowing for publication, I would suggest making some minor improvements:

- the abbreviation AWJ should be explained in the abstract,

- line 47 - I suggest deleting "Applications of NFRP:"

- line 69 - citing instead of link,

- line 131, 132 - units (dot - multiplication), the same in tables 3 and 4,

- line 158 - this is the Swiss standard, it should be in the references,

- line 211 - wood particles need to be shown in the figure with arrows,

- line 212 - cracks need to be shown on the figure with arrows,

- the work would be more interesting if it showed photos of conventional machining, such as saw cutting or milling,

- what method was used to produce the samples, is it commercial material ?

- I do not understand table 3, was there samples S5 - S16 ? what were the processing parameters for them ?

After completing the above mentioned comments, I recommend the paper for publication.

Author Response

Good morning,

Thank you very much for advice and comments.

  • the abbreviation AWJ should be explained in the abstract (abbreviation AWJ added in abstract)
  • line 47 - I suggest deleting "Applications of NFRP" (applications on NFRP erased)
  • line 69 - citing instead of link (fixed it)
  • line 131, 132 - units (dot - multiplication), the same in tables 3 and 4 (fixed it)
  • line 158 - this is the Swiss standard, it should be in the references (standard added in references)
  • line 211 - wood particles need to be shown in the figure with arrows (shown in picture)
  • line 212 - cracks need to be shown on the figure with arrows (shown in picture)
  • what method was used to produce the samples, is it commercial material? (Commercially purchased extruded profile)
  • I do not understand table 3, was there samples S5 - S16? what were the processing parameters for them? 20 samples with different processing parameters were realized (groups Q1, Q2, Q3, Q4, Q5 depends on required surface quality). Subsequent evaluation was realized on two selected groups – 1st with the highest surface quality – Q5 and 2nd with the lowest surface quality Q1. Group Q1 – samples S1-S4 and group Q5 samples S17-S20.

Best regards

Zuzana Mitalova